# Understanding how digital mental health interventions can be optimised to improve longer term sustainability: Findings from a causal mediation analysis of the CONEMO trials

Nadine Seward[1,2]*, Wen Wei Loh[3,4], Jaime Miranda[5,6,7], Franciso Diez-Canseco[8,5], Heloisa Garcia Claro[9], Paulo Rossi Menezes[10], Ivan Filipe de Almeida Lopes Fernandes[11], Ricardo Araya[2]

**1** Centre for Clinical Brain Sciences, College of Medicine and Veterinary Medicine, Edinburgh University, Edinburgh, United Kingdom, **2** Centre for Global Mental Health, Health Service and Population Research Department, Institute of Psychiatry, Psychology & Neuroscience, King's College London, London, United Kingdom, **3** Department of Data Analysis, Ghent University, Gent, Belgium, **4** Faculty Health, Medicine, and Life Sciences, Methodologie & Statistiek, Maastricht University, Maastricht, Netherlands, **5** CRONICAS Centre of Excellence in Chronic Diseases, Universidad Peruana Cayetano Heredia, Lima, Peru, **6** Sydney School of Public Health, Faculty of Medicine and Health, University of Sydney, School of Medicine, Camperdown, Australia, **7** Universidad Peruana Cayetano Heredia, Lima, Peru, **8** Universidad Científica del Sur, Lima, Peru, **9** State University of Campinas, Campinas, São Paulo, Brazil, **10** Faculdade de Medicina, Universidade de São Paulo, São Paulo, Brazil, **11** Centre for Engineering, Modelling and Applied Social Sciences, Federal University of ABC, São Bernardo do Campo, São Paulo, Brazil

* nseward@ed.ac.uk

## Abstract

Two CONEMO trials in Lima, Peru and São Paulo, Brazil evaluated a digital mental health intervention (DMHI) based on behavioural activation (BA), that demonstrated improved symptoms of depression at three, but not six-months. To understand how we can optimize features of the DMHI to improve outcomes for longer-term effectiveness and eventually scale-up, we investigated mediators through which the intervention improved symptoms of depression, separately for the two trials and then using a pooled dataset. Trial data included adults with depression (Patient Health Questionnaire – 9 (PHQ-9) score ≥10) and comorbid hypertension and/or diabetes. Interventional effects, a robust causal inference based approach to mediation, was used to decompose the total effect of DMHI on improved symptoms of depression (PHQ-9 scores improving by at least 50% between baseline and six months (yes/no)) into indirect effects via: understanding session content without difficulty; number of activities completed to improve levels of activation; and levels of BA captured using the Behavioural Activation for Depression Short-Form. Understanding the content of the sessions without difficulty mediated a 10% [0.10: Bias corrected 95% CI: 0.03 to 0.15] improvement in depression symptoms; completing activities mediated a 12% improvement [0.12: 0.01 to 0.23]; BA mediated a 2% [0.02: 0.01, 0.05] improvement. Our findings suggest that DMHI based on BA should consider developing strategies

**Data availability statement:** All relevant data are within the paper and its Supporting Information files.

**Funding:** The authors received no specific funding for this work.

**Competing interests:** The authors have declared that no competing interests exist.

to help participants complete activities they find enjoyable to improve levels of activation and maintain the effects of the intervention in the longer-term. To improve longer term outcomes, DMHI should also ensure that sessions are tailored to the characteristics of the intended population, so it is simple for patients to understand and therefore they are able to complete activities.

## Introduction

### Background

Comorbidity of depression with other physical chronic conditions are on the rise, particularly in low- and middle-income countries (LMICs) [1]. In Latin America, most of the disease burden is now explained by chronic diseases including depression, hypertension, and diabetes. Very often, depression amplifies the morbidity, and mortality associated with other chronic health conditions [2]. There is an urgent need for chronic disease management to include treatment for co-existing depression [3,4].

In addition to the above, there is also a large treatment gap for depression. In LMICs the proportion of people with severe mental health conditions who reported receiving any treatment was typically between 10–25%, compared to 50% or more in high-income countries [5]. One of the main reasons for this treatment gap is the shortage of specialised human resources and infrastructure such is the case in Latin American countries like Brazil and Peru [6].

### The CONEMO trials

CONEMO (*Control Emocional* in Spanish and Portuguese) is a digital mental health intervention (DMHI) developed and tested in LMICs settings that aimed to integrate the management of depression with physical chronic conditions including diabetes and/or hypertension [7]. CONEMO is a six-week intervention based on behavioural activation for depression that took place in Lima, Peru, and São Paulo, Brazil. Although at three-months after enrolment, CONEMO was effective in improving symptoms of depression by at least 50% in the intervention arm compared to the enhanced usual care arm in both trials, there was no evidence to support this effect at six-months after baseline assessments

Given the potential to bring this intervention to scale, it is important to understand how the CONEMO DMHI can be optimised to maximise benefits for this treatment in the longer-term. The CONEMO trials conducted in Brazil and Peru are particularly insightful to address this, as the DMHI captured multiple forms of information on the sessions that could shed light as to why the effectiveness measured at three months did not hold at six months [7].

The objective of our study is to use interventional indirect effects [8] to investigate mediators that are on the causal pathway between the CONEMO DMHI and the improvement of depressive symptoms at six-months by at least 50%. Understanding the mechanisms through which CONEMO improved symptoms of depression can provide insight into what features of the intervention can be targeted to improve

outcomes in the longer term. A priori we theorised that participants who were able to understand the content of the sessions without difficulty (M1), would be more likely to complete the activities they had self-selected to improve levels of enjoyment (M2), and that this, in turn, would improve levels of activation and avoidance (M3), and eventually improve their mood. Our theoretical model underpinning these hypotheses can be found in Fig 1. We will investigate these mediators separately for the different sites, then using a pooled dataset.

## Methods

### Setting

This is a secondary data analysis using data from the CONEMO trials that took place between September 2016 and April 2018 in twenty health facilities in São Paulo, Brazil and seven in Lima, Peru [7]. Data for this study was accessed in 01/Oct/2023. All data was anonymised where individual participants could not be identified.

### Design

The trial in São Paulo was a cluster randomised trial whilst the trial in Lima was an individually randomised trial. Participants in both trials were eligible if they were aged 21 or greater, had hypertension and/or diabetes, and clinically significant depressive symptoms (Patient Health Questionnaire-9 [PHQ-9] score ≥10) [7]. All participants gave written informed consent prior to entering the trial.

### CONEMO intervention

CONEMO is a low-intensity DMHI based on behavioural activation (BA) for depression and delivered through a smartphone (Trial Registration: ClinicalTrials.gov: NCT02846662 (São Paulo) and NCT03026426 (Lima)) [7]. Participants were recruited between 19/September/2016 and 2/April/2018 in Brazil and between 24/January/2017 and 30/March/2018 in Peru. The app consisted of 18 mini-sessions, delivered over a six-week period at a rate of three mini-sessions per week. Each session required fewer than 10 minutes to complete and focused increasing participation in pleasant or meaningful activities. Sessions also suggested activities to improve comorbid physical conditions (e.g., healthy diet). CONEMO was identical in both study sites, except it was supported by nurses in Lima and nurse assistants in São Paulo.

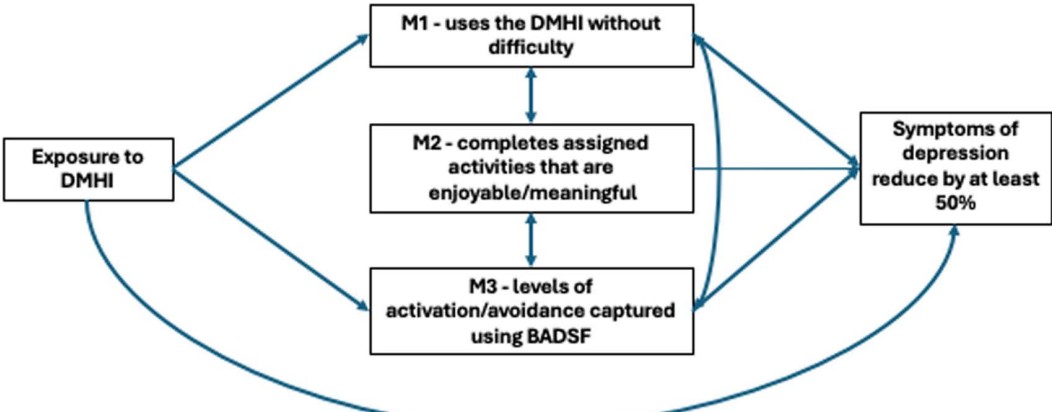

**Fig 1. Causal diagram demonstrating the proposed mediating pathways through which the CONEMO intervention may improve PHQ-9 scores in Lima and São Paulo at the six-month follow-up visit.** * The bi-directional arrows in Fig 1, e.g., between M1 and M2, express that no assumptions are made regarding the possible direction of a causal effect (e.g., between M1 and M2), or the presence of unmeasured common causes (e.g., of M1 and M2). * M1 measured at the three-month follow-up visit *M2 measured at the sessions via the DHMI.

At the start of the study, nurses/nurse assistants met with participants for an initial face-to-face meeting. At this point, participants received a smartphone with the preinstalled app and completed a tutorial on its use. Two additional phone calls were made to the participants at the beginning of the study to assist with any difficulties and to enhance motivation for using the digital intervention. Additional calls were prompted through notifications sent to nurses/nurse assistants when the automated system detected non-adherence.

Participants in the intervention arm received the DMHI. Participants in both study arms received enhanced usual care that consisted of being assessed for depression symptoms up to four times during the first month of the trial, and again during research follow-up assessments [7].

To assess whether any short-term (three-months) improvements were maintained over time, participants in both arms were assessed again at six-months. Details of the main trial methodology and results can be found elsewhere [7].

## Exposure

Participants in the intervention arm were offered the CONEMO DMHI and enhanced usual care (exposed to effects of the DMHI) whereas participants in the control arm were offered enhanced usual care only (unexposed).

## Outcome

The trials used the primary outcome of an improvement of at least 50% in PHQ-9 scores [9] between baseline and three-month assessments [7]. This indicator has been used in many depression treatment trials and is considered a robust way of ascertaining depression treatment improvements [10–12]. The PHQ-9 instrument also has good psychometric properties that has been used extensively in depression trials in LMICs [13]. As an example, in a similar context, Argentina, the PHQ-9 had high internal consistency (Cronbach's alpha = 0.87) [14]. For our mediation analysis, we used the outcome of improvement of at least 50% in PHQ-9 scores between baseline and the six-month follow-up visit. In doing so, we can establish temporality between the mediators and the outcome to rule out reverse causation, as our mediators were measured earlier at the three-month assessments.

## Mediators

A mediator is defined as an intervening variable on a causal pathway between an exposure and an outcome [15]. Causal mediation analysis using interventional (in)direct effects is grounded in a potential outcomes framework. This approach requires probabilities of the mediators to be simulated in both the exposed and unexposed (counterfactual) population using predictive models. For mediators measured in both arms of the trials, observed values in the experimental arm were used to simulate probabilities for the exposed population, and the observed values in the control arm were used to simulate probabilities for the unexposed population. For mediators measured in the experimental arm only, the values were categorised so that an "unexposed" value of zero was defined for those who were assigned to the experimental arm but were not exposed to the mediator of interest (i.e., did not complete any of the assigned activities; similar to those in the control arm who were unexposed). A full description of the potential mediators and selection criterion can be found in S1 File. The following mediators were selected.

**The messages on the DMHI are understood without difficulty (M1 – measured in the intervention arm only at the three-month follow-up visit).** Whether or not a participant understands the content of the messages delivered through the DMHI without difficulty, can influence not only whether a participant completes activities but also levels of activation and avoidance measured at three-months. We theorised that participants who found the DMHI difficult to use, would also have difficulty completing assigned activities, potentially not improving levels of activation and reducing levels of avoidance. In turn, this would influence whether participants experienced a reduction in symptoms of depression. At the three-month follow-up assessment (up to six weeks after sessions were completed), participants in the intervention arm were asked 'how much difficulty did you have understanding the messages on the DMHI' where they could respond

'0=very easy; 1=easy; 2=difficult; 3=very difficult'. As there were no differences between the 0 and 1 categories with the outcome these were grouped into one category. Likewise, as there were no differences between 2 and 3 categories with the outcome these were also grouped into one category. To capture this effect, we created a binary variable that ascertained whether the participants understood the content of the sessions delivered via the DMHI without difficulty (M1: no=unexposed; yes=exposed).

**Number of assigned activities on the DMHI completed (M2 – measured in the intervention arm only and captured automatically via the DMHI when participants completed a session).** At each session, the DMHI relayed messages that encouraged participants to select and complete a set of activities they found enjoyable. At the subsequent session, participants indicated via a questionnaire in the DMHI if they completed the activity. We theorised that participants who completed activities that were self-selected to improve levels of enjoyment, would help to improve levels of activation and reduce levels of avoidance, potentially reducing symptoms of depression. M2 captures the number of activities that a participant reports completing (0–27). We categorised this mediator into the following: no assigned activities completed (unexposed); 1–10 assigned activities completed (exposed); and 11–27 assigned activities completed (exposed). Our first categorization was based on being completely unexposed to this mediator (completing no sessions) that is required for counterfactual based approach to mediation. However, subsequent categorisations were based on associations with the outcome of reduction in symptoms of depression by at least 50%.

**Level of behavioural activation (M3 – measured in both trial arms at three months).** CONEMO was based on the provision of BA techniques that investigators theorised would lead to increased activation (and decreased avoidance). Structuring activities that participants find enjoyable and meaningful will increase contact with positive reinforcement for nondepressed behaviours and increased positive mood [16]. Over time this process will lead to decreased depressive symptoms. The Behavioural Activation for Depression Scale Short Form (BADS-SF) was designed to assess the levels of activation and avoidance. We theorised that participants who completed a higher proportion of activities they found enjoyable and meaningful (M2) would have higher levels of activation and avoidance captures with the BAD-SF instrument (M3). The BAD-SF and has good psychometric properties with high internal consistency (Cronbach's alpha=0.82) [17]. Levels of behavioural activation for the CONEMO DMHI was therefore captured using the BAD-SF in both intervention (exposed) and control arms (unexposed) [17]. BADS-SF scores range from zero to 54, with higher scores indicating higher levels of behavioural activation.

**Dependency of mediators on one another.** A by-product of the interventional indirect effects is that in addition to the mediator-specific indirect effects, there is another indirect effect via the mediators' mutual dependence on one another [8]. This indirect effect is close to zero when mediators are conditionally independent and non-zero when mediators interact or covary in their effects on the outcome.

### Mediator outcome confounders

We considered more than thirty potential baseline characteristics that are not influenced by the intervention, as potential confounders. Any potential mediator-outcome confounder was included if it was associated with the mediator or the outcome ($p < 0.10$). Models for the pooled dataset included any mediator-outcome confounder included in either of the individual sites as well as a variable to represent study site. A detailed description of selecting our mediator-outcome confounders can be found in S1 File.

### Statistical analysis

**General.** To better understand how the DMHI influenced the mediators to improve symptoms of depression, we compare exposed and unexposed levels of the mediators (please see description, of mediators for exact definitions of exposure status) with both exposure to the intervention (control vs intervention arm) and the outcome of improved symptoms of depressions (a PHQ-9 score reduction of at least 50% between baseline and six-months (yes/no))

separately for the different trials as well as using the pooled dataset. For each comparison, we also provide descriptive statistics in baseline characteristics selected as our mediator-outcome confounders.

**Mediation analysis.** We aimed to investigate the extent to which improved symptoms of depression at six-months after baseline, captured using the PHQ-9 questionnaire, was explained via selected mediators. To achieve this, we used the interventional indirect effects approach for causal mediation analysis to understand population level effects relevant to this analysis [8]. Interventional effects provides a robust causal inference framework that allows to simultaneously evaluation for multiple mediators, their dependences, interactions, and non-linearities. For instance, this approach allowed us to test for moderated mediation (i.e., the indirect effect is influenced by a baseline variable) as well as the moderating effect of baseline variables on the association between exposure to the intervention and the outcome. Importantly, this approach allows mediators to interact with one another. This influences not only the indirect effects for all the mediators, but also the direct and total causal effects.

We applied this approach to mediation separately for the different CONEMO trials, then using the pooled dataset. The final model that was considered for this mediation analysis in both trials is shown in Fig 1. The mediation analysis conformed to AGReMA statement for reporting mediation analysis of randomised trials [15]. Further details on the mediation analysis relating to the interventional effects can be found in S1 File.

**Estimation and model fit.** Estimation for the interventional indirect effects was based on Monte Carlo integration using a 1,000-fold expanded dataset [8]. The expanded dataset was created in four steps separately for each of the different sites. Details of each step, and details of the models including interactions and selected confounders can be found in S1 File. Bias-corrected confidence intervals were based on nonparametric bootstrap with 1,000 resamples that adjusted for clustering and stratification [8].

**Assumptions.** Due to the randomised nature of the CONEMO trials, the assumptions about no unmeasured confounders between the exposure and each mediator, and between the exposure and the outcome, are fulfilled. The main assumption relevant to our study is that there is no unmeasured mediator-outcome confounding.

**Missing data.** In São Paulo out of the 880 participants enrolled at baseline, 656 (85%) had data available for the mediation analyses. In Lima, out of 432 participants enrolled at baseline, 389 (90%) had data available for the mediation analysis. Due to the non-random nature of the missing data, we were not able implement the imputation models (S2 File). Analyses were therefore based on complete data only.

**Sensitivity analyses.** Primary outcomes for depression are mainly reported as either recovery from depression (PHQ-9 < 10) or an improvement of at least 50% in PHQ-9 scores between baseline and follow-up. We selected our outcome measure of improved symptoms of depression (50% reduction in PHQ-9 scores between baseline and the six month follow-up) to ensure consistency with the outcomes reported in the main trial [7]. To facilitate comparability with other studies, we also conducted the same analyses using recovery from depression at six-months (PHQ-9 < 10).

## Ethics and consent statement

The authors assert that all procedures contributing to this work comply with the ethical standards of the relevant national and institutional committees on human experimentation and with the Helsinki Declaration of 1975, as revised in 2013. All procedures involving human subjects/patients were approved by Institutional Review Board at the University of São Paulo (nº 457.605) and the National Commission of Ethics in Research (nº 355.039). In Lima, the protocol was approved by the Institutional Review Board at the Universidad Peruana Cayetano Heredia (nº 34516–16).

## Results

### General

Table 1 demonstrates that without adjustment for mediator-outcome confounders, participants in the intervention arm were more likely to have higher levels of activation and avoidance captured using the BADS-SF (M3) compared to participants

**Table 1. Comparison of exposed and unexposed levels of the mediators captured at three months and baseline mediator-outcome confounders, with exposure to the intervention using the pooled data, and separately for the different study sites.**

| Mediators captured at three months | Lima, Peru (n = 389) | | São Paulo, Brazil (n = 656) | | Pooled data (n = 1045) | |
|---|---|---|---|---|---|---|
| | Control arm (n = 192, 49.4%) | Intervention arm (197, 50.6%) | Control arm (n = 343, 52.3%) | Intervention arm (n = 313, 47.7%) | Control arm (n = 535, 51.2%) | Intervention arm (n = 510, 48.8%) |
| M1: numbers (proportion) of participants who understood content of sessions without difficulty | a | 150 (76.1) | a | 245 (78.3) | a | 395 (77.5) |
| M2: numbers (proportion) of assigned activities completed | | | | | | |
| No activities completed | a | 8 (4.1) | a | 54 (17.3) | a | 62 (12.2) |
| 1-10 activities completed | | 166 (84.3) | | 161 (51.4) | | 327 (64.1) |
| 11-27 activities completed | | 23 (11.7) | | 98 (31.3) | | 121 (23.7) |
| M3: levels of activation and avoidance captured with the BAD-SF instrument (Mean, SD) | 25.9 (8.9) | 29.8 (9.0) | 25.1 (8.9) | 26.2 (8.5) | 25.4 (8.9) | 27.6 (8.8) |
| Mediator outcome confounders captured as baseline | | | | | | |
| PHQ-9 (Mean, SD) | 14.1 (3.8) | 14.1 (3.8) | 15.8 (4.2) | 16.1 (4.4) | 15.2 (4.1) | 15.3 (4.3) |
| Age (Mean, SD) | 59.0 (11.8) | 60.5 (10.2) | 55.1 (11.7) | 54.7 (11.2) | 56.8 (11.8) | 56.9 (11.2) |
| Years of education (Mean, SD) | 10.1 (3.7) | 10 (3.9) | 7.8 (3.9) | 7.9 (3.8) | 8.7 (4.0) | 8.7 (4.0) |
| Numbers (proportion) of participants taking medication to treat emotional problem | 21 (10.9) | 23 (11.7) | 126 (36.7) | 103 (32.9) | 147 (27.5) | 126 (24.7) |
| Numbers (proportion) of participants who have friends to contact if needed | 149 (77.6) | 149 (75.6) | 289 (84.3) | 270 (86.3) | 438 (81.9) | 419 (82.2) |
| Number of everyday activities mobile phone used for (Mean, SD) [b] | 4.4 (3.5) | 3.8 (3.0) | 4.5 (3.4) | 4.9 (3.3) | 4.4 (3.4) | 4.5 (3.2) |
| Levels of activation and avoidance captured using the BAD-SF instrument (Mean, SD) | 25.8 (7.8) | 25.5 (7.7) | 23.3 (8.2) | 22.6 (8.3) | 24.2 (8.1) | 23.7 (8.2) |

a. Data relevant only if received intervention and therefore collected in the intervention arm only.

b. Participants were asked 10 questions on mobile phone use. A higher score reflects a greater proficiency in using mobile phones.

in the control arm at the different study sites, as well as using the pooled data. Mediator-outcome confounders were generally equally distributed between the intervention and control arms.

Descriptive statistics presented in Table 2 demonstrates that participants who found the DMHI easy to use (M1), completed assigned activities (M2), and had higher levels of activation and avoidance (M3), were more likely to experience a 50% reduction in symptoms of depression. Use of medication to treat an emotional problem was associated with a reduction in symptoms of depression improving at six months in the pooled dataset and at each study site. Higher levels of activation and avoidance at baseline were also associated with symptom reduction at six months.

## Mediation analyses

**Moderated mediation.** Using the pooled dataset as well as the São Paulo dataset, there was some evidence of moderated mediation whereby medication use for emotional problems at baseline, moderated levels of activation and avoidance (M3) at three months, to improve the outcomes at six months ($p = 0.001$). Specifically, participants taking medication and experiencing lower levels of activation and avoidance (M3), had a reduced probability in symptoms of depression reducing by 40%, compared to participants who were not taking medication. Findings using data from the pooled dataset also indicated that study site moderated levels of activation and avoidance (M3), to improve outcomes ($p = 0.032$). For example, participants in Lima with lower levels of activation and avoidance (M3), had a higher probability of symptoms of depression reducing by 40%, compared to participants in São Paulo.

**Table 2. Comparison of exposed and unexposed levels of the mediators captured at three months, with the outcome of improved symptoms of depression (a reduction in PHQ-9 scores of at least 50% at six-month from PHQ-9 scores at baseline) using the pooled data, and separately for the different study sites.**

| Mediators captured at three months | Lima, Peru (n = 389) | | São Paulo, Brazil (n = 656) | | Pooled data (n = 1045) | |
|---|---|---|---|---|---|---|
| | No symptom reduction (n = 181, 46.5%) | Symptom reduction (n = 208, 53.5%) | No symptom reduction (n = 371, 56.6%) | Symptom reduction (n = 285, 43.5%) | No symptom reduction (n = 552, 52.8%) | Symptom reduction (n = 493, 47.2%) |
| M1: numbers (proportion) of participants who understood content of sessions without difficulty | 60 (68.2) | 90 (82.6) | 117 (31.5) | 128 (44.9) | 177 (38.6) | 218 (55.3) |
| M2: numbers (proportion) of assigned activities completed | | | | | | |
| No activities completed | 5 (5.7) | 3 (2.8) | 39 (23.4) | 15 (10.3) | 44 (17.3) | 18 (7.1) |
| 1-10 activities completed | 75 (85.2) | 91 (83.5) | 83 (49.7) | 78 (53.4) | 158 (62.0) | 169 (66.3) |
| 11-27 activities completed | 8 (9.1) | 15 (13.8) | 45 (27.0) | 53 (36.3) | 53 (20.8) | 68 (26.7) |
| M3: levels of activation and avoidance captured with the BAD-SF instrument (Mean, SD) | 26.2 (8.8) | 29.4 (9.2) | 23.8 (8.2) | 28.0 (8.8) | 24.6 (8.4) | 28.6 (9.0) |
| Mediator outcome confounders captured as baseline | | | | | | |
| PHQ-9 (Mean, SD) | 13.3 (3.3) | 14.5 (4.1) | 16.1 (4.3) | 15.8 (4.3) | 15.2 (4.2) | 15.4 (4.2) |
| Age (Mean, SD) | 60.2 (11.2) | 59.4 (10.9) | 55.5 (10.8) | 54.7 (12.3) | 57.0 (11.1) | 56.7 (11.9) |
| Years of education (Mean, SD) | 7.9 (3.9) | 7.8 (3.9) | 10.2 (4.0) | 9.9 (3.7) | 8.7 (4.0) | 8.7 (3.9) |
| Numbers (proportion) of participants taking medication to treat emotional problem | 22 (12.2) | 22 (10.6) | 164 (44.2) | 65 (22.8) | 186 (33.7) | 87 (17.7) |
| Numbers (proportion) of participants who have friends to contact if needed | 141 (77.9) | 157 (75.5) | 313 (84.4) | 246 (86.3) | 454 (82.3) | 403 (81.4) |
| Number of everyday activities mobile phone used for (Mean, SD) [a] | 4.1 (3.3) | 4.0 (3.2) | 4.6 (3.3) | 4.8 (3.4) | 4.4 (3.3) | 4.5 (3.3) |
| Levels of activation and avoidance captured using the BAD-SF instrument (Mean, SD) | 25.0 (7.4) | 26.2 (8.0) | 21.8 (7.7) | 24.4 (8.7) | 22.9 (7.7) | 25.1 (8.4) |

a. Participants were asked 10 questions on mobile phone use. A higher score reflects a greater proficiency in using mobile phones.

**Interventional total causal effect and indirect effects.** Results displayed in Table 3, demonstrated that at six-months after baseline, there was some evidence to support a small difference in the probability of symptoms of depression improving by at least 50% (yes/no) between the intervention and control arm using data from the pooled dataset (0.04 [95% bias-correct confidence interval: 0.00 to 0.09]), but not in the separate trials in Lima and São Paulo (Table 3).

Finding the content of the sessions easy to understand (M1) mediated a 10% difference in the probability of symptoms of depression improving by at least 50% (yes/no) in the pooled dataset (difference in probability of symptoms of depression improving by at least 50% between participants who understood the content of the sessions without difficulty, compared to those who found the content of the sessions difficult to understand: 0.10 [95% bias-corrected confidence interval: 0.03 to 0.15]), as well as in Lima (0.11 [0.02 to 0.26]) and São Paulo (0.09 [0.02 to 0.15]) separately.

Estimates using data from the pooled dataset provide evidence that completing the activities as intended (M2) was responsible for mediating a 12% difference in the probability of symptoms of depression improving by at least 50% (mean difference in symptoms of depression improving by at least 50% between participants who completed assigned activities

PLOS Global Public Health

**Table 3. Total effect and interventional (in)direct effects of the CONEMO intervention on improved symptoms of depressions measured depression (a reduction in PHQ-9 scores of at least 50% at six-month from PHQ-9 scores at baseline) at six-months follow-up.**

| Effect | Pooled dataset (n = 1045) | Lima (n = 389) | São Paulo (n = 656) |
|---|---|---|---|
| | Estimates (bias-corrected 95% CI)[a,b,c,] | Estimates (bias-corrected 95% CI)[a,b,c,] | Estimates (bias-corrected 95% CI)[a,b,c,d] |
| Total effect of the DMHI on improved symptoms of depression | 0.04 (0.00, 0.09) | 0.04 (-0.04, 0.12) | 0.04 (-0.03, 0.10) |
| Direct effect | -0.22 (-0.32, -0.10) | -0.15 (-0.37, 0.23) | -0.21 (-0.32, -0.08) |
| Indirect effect of understanding content of sessions without difficulty (M1) | 0.10 (0.03, 0.15) | 0.11, (0.02, 0.26) | 0.09 (0.02, 0.15) |
| Indirect effect of completing assigned activities (M2) | 0.12 (0.02, 0.23) | 0.04 (-0.32, 0.30) | 0.12 (-0.01, 0.24) |
| Indirect effect of behavioural activation (M3) | 0.02 (0.01, 0.05) | 0.04 (0.00, 0.07) | 0.02 (0.01, 0.04) |
| Indirect effect through the dependence of mediators on one another [e] | 0.03 (0.01, 0.07) | -0.01 (-0.03 0.02) | 0.02 (0.01, 0.05) |

[a] Estimates have been adjusted for mediator-outcome confounders of baseline medication, age, mobile use, baseline PHQ-9 scores, and education in São Paulo; and baseline medication, age, mobile use, baseline PHQ-9 scores, and education, and support from colleagues in Lima. [b] Estimation for the different effects was based on Monte Carlo integration using 1,000-fold expanded dataset. [c] Bias-corrected confidence intervals were based on nonparametric bootstrap with 1000 resamples adjusting for clustering.

compared to those who did not complete any assigned activities: 0.12 [0.02 to 0.23]). There was some evidence to support this finding in São Paulo (0.12 [-0.01 to 0.24), however there was no evidence to support the same finding in Lima (0.04 [-0.32 to 0.30]).

There was evidence that behavioural activation (M3) mediated a 2% difference in the probability of symptoms of depression improving by at least 50% using the pooled dataset (mean difference in proportion with improved symptoms of depression between participants with BAD-SF scores in the exposed population, compared to participants with BAD-SF scores in the unexposed population: 0.02 [95% bias-corrected confidence interval: 0.01 to 0.05]), as well as in São Paulo (0.02 [0.01 to 0.04]), and in Lima (0.04 [0.00, 0.07]).

Lastly, using the pooled dataset, the indirect effect attributable to the mutual dependence of the mediators on one another, indicate there was a 3% difference in symptoms of depression improving (0.03 [95% confidence interval: 0.01 to 0.07]), and a 2%, difference in São Paulo (0.02 [0.01, 0.05]. There was no evidence to support this in Lima (-0.01 [-0.03, 0.02]).

## Sensitivity analyses

Estimates from our analyses using as outcome the probability of recovery from depression between the intervention and control arm at six-months (defined as PHQ-9 score <10 at the sixth month follow-up visit), were largely consistent with the findings using the outcome measure of a reduction in PHQ-9 scores of at least 50% between baseline and the sixth month follow-up. Findings are reported in S2 File.

## Discussion

This novel application of the causal mediation framework of interventional indirect effects provides insight into how the CONEMO DMHI can be adapted to achieve better longer-term improvements in depressive symptoms. Whilst simultaneously accounting for all mediators, we have been able to successfully demonstrate the importance of ensuring the content of the app sessions is understood without difficulty. Importantly, the indirect effects of the number of assigned activities completed that were selected to improve levels of activation (M2) and activation levels captured using the BADS-SF

(M3), confirm the importance of targeting behavioural activation to improve depression outcomes in the longer term. Our findings, benefit from the evaluation of the same DMHI in two different contexts, thus allowing for the assessment of how mediators operate in diverse real-world settings.

A priori, we theorised that understanding session content without difficulty (M1) will influence the number of activities completed that were selected to improve behavioural activation (M2), and therefore activation and avoidance levels (M3). The first two mediators are necessary pre-conditions required to maximise the benefits of the DMHI, that was based on behavioural activation for depression. This theory was supported by our findings that indicated the mutual dependence of the mediators on one another, mediated an improvement in our outcome.

Findings from both study sites and the pooled analyses suggest that the intervention was less effective at six-months among those participants who found the content of the sessions difficult to understand. Adapting the DMHI so that the sessions are understood without difficulty by all participants (including those with low levels of digital literacy and/or education), can also help to ensure that more activities are completed, and levels of behavioural activation are higher. Our findings that both the number of activities completed (M2) as well as activation and avoidance levels (M3) mediated improvements in depression outcomes, confirm the importance of psychological interventions targeting behavioural activation. It is important to note that M2 and M3 capture different properties of behavioural activation. Whereas the number of activities completed, is an indicator as to the dose-response of the intervention that targets activation, the BADS-SF captures levels of activation after the activities were completed as measured by the questionnaire that also includes items relating to avoidance.

Findings from this mediation analysis are also supported by qualitative research arising for the CONEMO trial in Lima that explored participants and nurses' experiences in using the DMHI. Some recommendations include making the DMHI more user-friendly by writing session content in simpler text and using more videos. These adaptations could potentially reduce the role of nurses employed on a full-time basis in Lima, that is not feasible if brought to scale [18].

There has been conflicting evidence to support the role of behavioural activation in mediating the effects of psychological therapies in sessions that are delivered both face to face and through digital mental health interventions [19,20]. However, many of the studies are heterogeneous with differences in the content of sessions, levels of depression, taking place in diverse contexts, often coupled with poor methodological quality making comparisons difficult [20–23]. More recently, a mediation analysis using data from a trial in Goa, India, using the same approach as in this paper, demonstrated a significant role of behavioural activation in mediating a reduction in symptoms of depression, but not attending sessions or completing assigned activities [24,25]. Ensuring future mediation analyses use robust approaches to mediation, as suggested by the AGReMA statement, will help to better understand the mechanisms through which psychological interventions improve outcomes of depression [15].

## Strengths and limitations

Our approach to meditation has several strengths. Importantly, our mediation analyses allowed us to include multiple mediators, their interactions and non-linearities. Failing to simultaneously account for the different mediating pathways by excluding mediators could over or underestimate the indirect and direct effects [26]. Our analyses also had a vast array of data that was collected via the DMHI tool that provided information that allowed us to estimate the effects in a less biased approach than having a participant recall at each session whether activities were completed or not. These analyses benefit from a large sample size using data pooled from two trials testing the same intervention and uses a robust approach to causal mediation analysis to understand how a complex psychological therapy delivered using a DMHI, improved symptoms of depression.

There are also limitations to our analyses. When applying the interventional effects to a randomised trial, the main underlying assumption is that all important mediator-outcome confounders are accounted for [27]. Failing to do so, can also potentially bias all estimates including the direct and indirect effects [28]. Another issue is the possibility of correctly

specifying the models that use baseline predictors to set the mediators to a random subject-specific draw. This is especially the case when there are high dimensional data (multiple predictors and interactions) such as the case with the CONEMO trials. One of the more novel methods to mediation analysis uses a combination of machine learning and interventional effects that can overcome this limitation [29].

Given that our first mediator was collected up to a month and a half after the intervention took place, there is a chance of recall bias. For instance, participants who had severe symptoms of depression may feel this has to do with the DMHI and report that it was difficult to use, compared to someone with less severe symptoms of depression. This bias could increase the proportion of the total effect mediated through our first mediator. Future DMHI should consider capturing this information, electronically as soon as the session is complete. Lastly, our second mediator representing the number of activities completed was captured using a categorical variable (no activities, 1–10 activities, 11–27 activities). Categorising this mediator could have led to diluting the indirect effect and inflating the direct effect [8]. There is an issue with the second mediator, where our unexposed category had a limited number of participants, leading to wide confidence intervals. However, when we pooled the datasets, this limitation was largely overcome. Nevertheless, it is possible the indirect effect through number of activities completed may be underestimated.

Future interventions should consider findings from this analysis when designing digital interventions to treat depression. In particular, careful consideration needs to be given to ensure the instructions in the DMHI are easy to understand for the most vulnerable. Co-design approaches involving people with lived experience as well as participants who are more likely find the digital app difficult to use (e.g., older participants with severe symptoms of depression) could help to ensure session content is easy to understand. Adequate human support could also be used to answer queries so as not to discourage patients from participation as well as to encourage patients to complete all activities. Importantly, although human support has been identified as an key feature for digital interventions in improving clinical effectiveness [30]. Too much will inflate the costs of digital interventions, making them difficult to sustain in poorly resourced settings.

## Conclusions

The future of mHealth applications, facilitated by the gradual transition toward remote care in recent years arising during the COVID-19 pandemic, shows great promise to reduce existing large treatment gaps in mental health, especially with the rapid development of robust methodologies to understand how to adapt intervention to improve outcomes, such as adaptive interventions (or dynamic treatment regimens), causal machine learning [29,31] and robust implementation research [32].

## Supporting information

**S1 File. Details of mediation analyses. selection of mediators, mediator-outcome confounders; decomposition of the total effect of the CONEMO intervention into direct and indirect effects.**
(DOCX)

**S2 File. Sensitivity analyses comparing estimates from interventional indirect effects between the outcome of recovery from depression at six-months and reduction in PHQ-9 scores by 50% between baseline and six-months in the pooled analyses, the Lima trial, and the trial in São Paulo.**
(DOCX)

**S3 File. Codebook for dataset.**
(XLSX)

**S4 File. Dataset used for this analysis.**
(CSV)

## Acknowledgments

We are grateful to the associate editor and two anonymous reviewers for their valuable feedback, which significantly improved the manuscript. We would like to thank all of those involved in the original CONEMO trials, especially the trial participants. We would also like to acknowledge the funders of the original trial, the National Institute of Mental Health (U19MH098780)

## Author contributions

**Conceptualization:** Nadine Seward, Ricardo Araya.

**Data curation:** Nadine Seward.

**Formal analysis:** Nadine Seward.

**Methodology:** Nadine Seward, Wen Wei Loh.

**Supervision:** Ricardo Araya.

**Writing – original draft:** Nadine Seward.

**Writing – review & editing:** Nadine Seward, Wen Wei Loh, Jaime Miranda, Franciso Diez-Canseco, Heloisa Garcia Claro, Paulo Rossi Menezes, Ivan Filipe de Almeida Lopes Fernandes, Ricardo Araya.

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
