## [Decision Letter · Decision Letter 0]

PGPH-D-25-00349

Understanding how digital mental health interventions can be optimised to improve longer term sustainability: findings from a causal mediation analysis of the CONEMO trials

Dear Dr. Seward,

Thank you for submitting your manuscript to PLOS Global Public Health. After careful consideration, we feel that it has merit but does not fully meet PLOS Global Public Health’s publication criteria as it currently stands. Therefore, we invite you to submit a revised version of the manuscript that addresses the points raised during the review process.

We look forward to receiving your revised manuscript.

Kind regards,

Joel Msafiri Francis, MD, MS, PhD

Academic Editor

Journal Requirements:

1. We have amended your Competing Interest statement to comply with journal style. We kindly ask that you double check the statement and let us know if anything is incorrect. 2. Please provide separate figure files in .tif or .eps format. For more information about figure files please see our guidelines:  https://journals.plos.org/globalpublichealth/s/figures https://journals.plos.org/globalpublichealth/s/figures#loc-file-requirements 3. Your manuscript is missing the following sections: Results. Please ensure these are present, and in the correct order, and that any references to subheadings in your main text are correct. An outline of the required sections can be consulted in our submission guidelines here: https://journals.plos.org/globalpublichealth/s/submission-guidelines#loc-parts-of-a-submission 4. We have noticed that you have uploaded Supporting Information files, but you have not included a list of legends. Please add a full list of legends for your Supporting Information files after the references list.

Additional Editor Comments (if provided):

Reviewers' comments:

Reviewer's Responses to Questions

**Comments to the Author**

1. Does this manuscript meet PLOS Global Public Health’s publication criteria ? Is the manuscript technically sound, and do the data support the conclusions? The manuscript must describe methodologically and ethically rigorous research with conclusions that are appropriately drawn based on the data presented.

Reviewer #1: Yes

Reviewer #2: Yes

2. Has the statistical analysis been performed appropriately and rigorously?

Reviewer #1: Yes

Reviewer #2: Yes

3. Have the authors made all data underlying the findings in their manuscript fully available (please refer to the Data Availability Statement at the start of the manuscript PDF file)?

Reviewer #1: Yes

Reviewer #2: Yes

4. Is the manuscript presented in an intelligible fashion and written in standard English?

Reviewer #1: Yes

Reviewer #2: Yes

5. Review Comments to the Author

Reviewer #1: Important note: This review pertains only to ‘statistical aspects’ of the study and so ‘clinical aspects’ [like medical importance, relevance of the study, ‘clinical significance and implication(s)’ of the whole study, etc.] are to be evaluated [should be assessed] separately/independently. Further please note that any ‘statistical review’ is generally done under the assumption that study specific methodological [as well as execution] issues are perfectly taken care of by the investigator(s). This review is not an exception to that and so does not cover clinical aspects {however, seldom comments are made only if those issues are intimately / scientifically related & intermingle with ‘statistical aspects’ of the study}. Agreed that ‘statistical methods’ are used as just tools here, however, they are vital part of methodology [and so should be given due importance]. I look at the manuscript in/with statistical view point, other reviewer(s) look(s) at it with different angle so that in totality the review is very comprehensive. However, there should be efforts from authors side to improve (may be by taking clues from reviewer’s comments). Therefore, please do not limit the revision only (with respect) to comments made here.

COMMENTS: Although the manuscript is flawless (rather very good / excellent), I have a little different opinion/views/observations/concerns or rather questions & suggestions regarding very few issues which are given below:

Firstly, I noted that your ABSTRACT is well drafted (in my opinion), but is ‘assay type’. It is preferable [refer to item 1b of CONSORT checklist 2010: Structured summary of trial/study design, methods, results, and conclusions] to divide the ABSTRACT with small sections like ‘Objective(s)’, ‘Methods’, ‘Results’, ‘Conclusions’, etc. which is an accepted practice of most of the good/standard journals [including this one, though the ‘Guidelines to Authors’ may/did not specify an Abstract format, it is desirable]. It will definitely be more informative then, I guess, whatever the article type may be. Since separate paragraphs (presently) summarize the manuscript, you just have to give different headings to paragraphs in abstract [very easy indeed].

Secondly, please confirm the level of measurement of all the variables. Though few variables are continuous in appearance they are likely to yield data that are in [at the most] ‘ordinal’ level of measurement and not in ratio level of measurement for sure [for example: PHQ-9 or BAD-SF]. Then application of suitable non-parametric (or distribution free) test(s) is/are indicated/advisable [even if distribution may be ‘Gaussian’ (also called ‘normal’)]. Agreed that there is/are no non-parametric test(s)/technique(s) available to be used as alternative in all situation(s), but should be used whenever/wherever they are available. Therefore, in short use suitable non-parametric test(s)/technique(s) while dealing with data that are in ‘ordinal’ level of measurement even if [despite that] the distribution may be ‘Gaussian’.

For ordinal variables descriptive summary statistic such as (Mean, SD) may not be a correct representation. In case of few/At some-times ‘pooling’ of different samples may be questionable [for example: ‘number of assigned activities completed’ or ‘Years of education’ or ‘Medication to treat emotional problem’ or ‘Have friends to contact if needed’ as given in table-1].

As pointed out in ‘important note’ above “This review pertains only to ‘statistical aspects’ of the study and so ‘clinical aspects’ should be assessed separately/independently [one should carefully consider/look at the clinical implications of the study]. In my opinion, to make this article [more] acceptable (which is quite possible and easy), a small amount of re-vision (re-drafting) may be needed. Therefore, ‘Minor revision’ is recommended.

Reviewer #2: The manuscript is a thorough investigation on how digital mental health interventions (DMHIs) can potentially be optimized for long-term impact. It provides valuable insights into understanding the role of different mediators such as ease of understanding session content, activity completion, and behavioural activation in improving depression outcomes.

The paper explains the structure of the CONEMO intervention well, describing how the app sessions are delivered over six weeks and how mediators like the number of activities completed (M2) and behavioural activation (M3) are measured through established instruments like the BADS-SF. Additionally, the inclusion of both individual trial results and pooled analysis emphasizes the robustness of the findings across different settings (São Paulo and Lima) which is good for the generalizability of your conclusions.

A few areas could benefit from additional clarification or expansion:

Generally, check the citation style of the journal, it is ‘[ ]’ instead of ‘( )’ brackets.

Methods

Line 208 to 212: Consider providing justification for the categorisation of the mediators.

Line 212 to 257: Consider clarifying the theoretical reasoning behind choosing these mediators and how they are interrelated.

Discussion

You have appropriately acknowledged potential issues such as recall bias for the first mediator and the limitations in categorizing the activity completion variable.

Consider expanding the discussion on how these limitations may affect the interpretation of the results and possible strategies to mitigate these effects in future studies.

The finding suggests that ensuring clear, simple session content can lead to higher rates of activity completion and greater levels of behavioural activation, ultimately resulting in improved mental health outcomes. More discussion on how these insights might be applied practically to enhance digital interventions at scale could make the paper even more impactful.

6. PLOS authors have the option to publish the peer review history of their article (what does this mean? ). If published, this will include your full peer review and any attached files.

**Do you want your identity to be public for this peer review?** For information about this choice, including consent withdrawal, please see our Privacy Policy .

Reviewer #1: No

Reviewer #2: No

---

## [Editor Report · Decision Letter 1]

Understanding how digital mental health interventions can be optimised to improve longer term sustainability: findings from a causal mediation analysis of the CONEMO trials

PGPH-D-25-00349R1

Dear Dr. Seward,

We are pleased to inform you that your manuscript 'Understanding how digital mental health interventions can be optimised to improve longer term sustainability: findings from a causal mediation analysis of the CONEMO trials' has been provisionally accepted for publication in PLOS Global Public Health.

Best regards,

Joel Msafiri Francis, MD, MS, PhD

Academic Editor